# Unpacking Psychological Vulnerabilities in Deaths of Despair

**DOI:** 10.3390/ijerph20156480

**Published:** 2023-07-31

**Authors:** Jieun Song, Sohyun Kang, Carol D. Ryff

**Affiliations:** Institute on Aging, 2245 Medical Sciences Center, University of Wisconsin-Madison, 1300 University Avenue, Madison, WI 53706, USA; kkang9@wisc.edu (S.K.); cryff@wisc.edu (C.D.R.)

**Keywords:** deaths of despair, deaths due to heart disease or cancer, eudaimonic well-being, hedonic well-being, educational status

## Abstract

Recent demographic findings show increased rates of death due to suicide, drug addictions, and alcoholism among midlife white adults of lower socioeconomic status (SES). These have been described as “deaths of despair” though little research has directly assessed psychological vulnerabilities. This study used longitudinal data from the Midlife in the U.S. (MIDUS) study to investigate whether low levels of eudaimonic and hedonic well-being predict increased risk of deaths of despair compared to other leading causes of death (cancer, heart disease). The investigation focused on 695 reported deaths with cause of death information obtained from 2004 to 2022 via NDI Plus. Key questions were whether risk for deaths due to despair (suicide, drug addiction, alcoholism) compared to deaths due to cancer or heart disease were differentially predicted by deficiencies in well-being, after adjusting for sociodemographic variables. Low levels of purpose in life, positive relations with others, personal growth and positive affect predicted significantly greater likelihood of deaths of despair compared to deaths due to heart disease, with such patterns prominent among better-educated adults. The findings bring attention to ongoing intervention efforts to improve psychological well-being.

## 1. Introduction

In the latter decades of the past century, the U.S. saw marked declines in mortality rates among midlife and older adults. These were generally seen as resulting from better health practices, increased prevention, and treatment efforts [1]. Between 1999 and 2013, however, marked increases in all-cause mortality became evident among middle-aged non-Hispanic whites [2]. The increases were primarily about deaths due to suicide, drug addiction, and alcoholism. These problems were concentrated among those of low socioeconomic status, although later work showed that rising premature mortality was not limited to middle-aged, uneducated whites [3,4]. The narrative that grew up around these findings was referred to as “deaths of despair” [5]. Limited, if any, assessment of despair was part of these inquiries. Rather, despair was inferred from the kinds of death examined. 

Subsequent work from the MIDUS (Midlife in the U.S.) national longitudinal study examined mental health from the mid-1990s to the early 2010s [6] and documented significant decline over time concentrated among low-SES adults. The largest changes were evident in reports of hedonic well-being, i.e., negative affect strongly increased over time, while positive affect and life satisfaction strongly decreased over time. Findings for a composite of eudaimonic well-being showed smaller decreases over time and a weaker differential by SES. This inquiry underscored that the observed patterns in mental health decline were evident across a broad range of ages from younger to older adults, and thus were not restricted to midlife. Other work used data from the National Health Interview Surveys to track age, period, and cohort effects in psychological distress [7]. Findings showed higher distress among more recently born birth cohorts. 

The evidence for increased midlife mortality unfolding over time, combined with heightened psychological vulnerabilities has been framed within a context of increased societal ills, including the opioid epidemic [8], heightened prevalence of alcohol use and related disorders [9], growing economic inequality [10], and decline of labors’ share in industrial profits [11]. Thus, across disciplines, ever-widening inequality is now seen as a major scientific imperative. A review of multidisciplinary findings from MIDUS [12] showed that, despite recent gains in educational attainment at the population level, the later refresher sample (recruited in 2012) had worse economic profiles, poorer physical health, and less well-being than the baseline core sample (recruited in 1995/1996). Additional findings showed growing inequality in physical health, mental health, and biomarkers. Some of these studies documented psychological factors as moderators or mediators of health inequality, for example, that inflammatory markers were heightened among low-education, minority adults who also had high profiles of anger [13,14].

To these prior endeavors, we bring three observations. First, despite growing interest in deaths of despair, limited research to date has assessed what kind of psychological vulnerabilities contribute to increased risk of death due to suicide, drug addiction, and alcoholism [15]. This issue elevates the prominence of psychological factors, both as predictors of increased risk of particular types of death, but also as potential targets for intervention. Second, amidst interest in deaths of despair, it is important to recognize that they constitute a relatively small fraction of overall deaths each year. For example, of the over 2.8 million U.S. deaths in 2019 [16], the leading cause of death was heart disease (23%), followed by cancer (21%); notably fewer deaths were attributed to suicide, drug addiction, and alcoholism (7%). A central question is whether psychological vulnerabilities are restricted only to deaths of despair, as the narrative implies, or are evident across other major causes of death as well. 

A major question is what constitutes relevant indicators of despair? Shanahan [17] argue that psychiatry is well positioned to define and measure despair, which they suggest includes cognitive, emotional, behavioral, and biological components [18]. This wide sweep may, however, blur boundaries between mental health per se and the factors that shape or follow from psychological experience. Thus, we build on Goldman, Glei and Weinstein [6] who acknowledged there were no established formulations of despair, or consensus about how to measure it, but chose to examine widely used measures of eudaimonic and hedonic well-being. Their findings, as noted above, showed cross-time decline in these positive aspects of mental health. 

The present inquiry examines a specific dimension of eudaimonia, namely, purpose in life, given the focus of this Special Issue of IJERPH. This aspect of well-being emerged from Victor Frankl’s [19] writings about surviving Nazi concentration camps. During his three-year ordeal, he came to view purpose in life as a sustaining force in the face of extreme trauma. After his release, he wrote about his insights from the experience and subsequently developed logotherapy as a way of helping individuals see or create a sense of purpose vis à vis the existential challenges of life. Low purpose in life, in contrast, may constitute a notable vulnerability factor in the face of notable life challenge. 

Interest in eudaimonic well-being and its relevance for health has grown rapidly in recent years [20,21]. As such, we investigate whether deficiencies in multiple aspects of positive functioning from the original model [22] might be relevant markers of vulnerability that increase risk of certain kinds of death. Specifically, we considered whether low levels of autonomy, environmental mastery, personal growth, positive relations with others, purpose in life, or self-acceptance increase risk for deaths of despair compared to risk for death due to heart disease or cancer. The absence of hedonic well-being, such as a paucity of positive affect or life satisfaction, or high levels of negative affect may also create vulnerability for deaths of despair. Thus, our analyses included multiple indicators of eudaimonic and hedonic well-being. 

The inquiry is situated amidst growing findings that link purpose in life to better health and extended longevity. Multiple studies document that high purpose in life predicts reduced risk of mortality [23] as well as reduced risk for Alzheimer’s disease, cardiovascular disease, and stroke [24,25,26,27], after adjusting for numerous covariates. High purpose in life also predicts better health care practices, including better sleep, as well as reduced risk of future drug misuse [28,29,30]. Our inquiry shifts the focus to the other end of the distribution—namely, whether low levels of purpose in life heighten risk for death due to suicide, drug addiction or alcoholism. We further broaden the inquiry to include a comprehensive battery of eudaimonic and hedonic well-being. 

In sum, the key objectives of the current study were to examine emerging mortality data from the MIDUS national longitudinal study to assess risk for deaths of despair (due to suicide, drug addiction, alcoholism) relative to other major causes of death (heart disease, cancer). Diverse psychological vulnerabilities were considered as risk factors, formulated as deficiencies in numerous aspects of eudaimonic and hedonic well-being. Whether the obtained patterns varied by the educational status or gender of participants was also considered. 

## 2. Materials and Methods

### 2.1. Data and Sample

We analyzed data from the MIDUS study, a longitudinal survey of a national probability sample of non-institutionalized, English-speaking adults who were age 25 to 74 during the initial survey in 1995–1996 (MIDUS 1, *n* = 7108). Respondents were surveyed again in 2004–2006 when they were aged 35 to 84 (MIDUS 2, *n* = 4963). To increase the inclusion of Blacks, MIDUS 2 was expanded to include a stratified sample of Blacks from Milwaukee, WI, via area probability sampling method (*n* = 592) [31]. The MIDUS 2 participants were evaluated again in 2013–2014 (MIDUS 3, *n* = 3683). 

The mortality data cover the deaths that occurred by the end of 2022. The analytic sample consists of MIDUS participants who (1) completed interview and self-administered questionnaire (SAQ) at MIDUS 2 and (2) died after MIDUS 2 and the cause of death information was available via NDI Plus (2004–2022). NDI Plus (National Death Index Plus) provides the cause of death information in addition to basic NDI information (NDI User’s Guide: https://www/cdc.gov/nchs/data/ndi/ndi_user_guide.pdf (accessed on 1 February 2023).

Of 4457 MIDUS 2 respondents who completed interview and SAQ, 1271 were ascertained as decedents via search of NDI files by 2022 and 1252 had valid information of cause of death including underlying and multiple cause of death. The analytic sample consisted of the decedents who died of despair related causes (suicide, drug addiction, alcoholism) (*n* = 168; 94 men, 74 women), cancer (*n* = 256; 127 men, 129 women), or heart disease (*n* = 271; 145 men, 126 women). 

### 2.2. Measures

*Cause of death: deaths of despair, heart disease, cancer.* Membership in the cause of death groups was based on two types of information from NDI Plus (2004–2022): underlying cause of death and multiple cause of death. Decedents were placed in the “death of despair” group if their cause of death in NDI Plus, in either the multiple cause of death or underlying cause of death measures, included suicide, drug addictions, or alcoholism including alcoholic liver diseases. 

*Eudaimonic well-being.* Six components of psychological well-being scale [22] were assessed, including purpose in life (e.g., “I have a sense of direction and purpose in life”), positive relations with others (e.g., “Most people see me as loving and affectionate”), personal growth (e.g., “I think it is important to have new experiences that challenge how you think about yourself and the world”), environmental mastery (e.g., “I am quite good at managing the many responsibilities of my daily life”), self-acceptance (e.g., “In general, I feel confident and positive about myself”), and autonomy (e.g., “I have confidence in my opinions, even if they are contrary to the general consensus”). Each component includes seven items measured on a 7-point scale ranging from strongly agree to strongly disagree. The score was calculated by summing each item’s score and high scores indicated higher levels of each component. 

*Hedonic well-being*. Positive affect was measured with 4-items adopted from the PANAS (Positive and Negative Affect Schedule [32]) for MIDUS. The questions asked how much of the time the participant felt enthusiastic [attentive, proud, active], during the past 30 days (1 = all of the time to 5 = none of the time). The measure of negative affect consists of 5-items adopted from the PANAS for MIDUS. They asked how much of the time the participant felt afraid [jittery, irritable, ashamed, upset], during the past 30 days (1 = all of the time to 5 = none of the time). The third measure assessed life satisfaction with a mean score of five items asking the participants’ rating of their life overall, work, health, relationship with spouse/partner, and relationship with children (0 = the worst possible to 10 = the best possible). 

*Education: no college degree* vs. *Bachelor’s degree or more*. Education level was assessed by the highest level of education completed at MIDUS 2 (no college degree [e.g., high school graduation, some college education without degree] vs. Bachelor’s degree or more). Prior work has sometimes used three levels of educational status [33], but the limited number of deaths precluded this more differentiated assessment, which could yield unstable results. In addition, findings from 1999 to 2019 showed that deaths of despair were concentrated among those with less than a college education [2,4]. 

*Covariates*. Several variables from prior research found to be associated with the prevalence of specific cause of death were included as covariates, including age (in years), gender, race/ethnicity (non-Hispanic white vs. others), household income (in 2005 dollars, log transformed), marital status (currently married vs. unmarried [divorced, widowed, never married]), and employment status (currently working vs. not working). Select health behaviors, binge drinking (having five or more drinks at one occasion) and smoking, were also included as possibly relevant factors or pathways through which deaths of despair occur. 

### 2.3. Analysis Plan

Sociodemographic characteristics (age, gender, race/ethnicity, education, income, marital status, and employment status) and psychological vulnerabilities (low eudaimonic well-being, low hedonic well-being) of the three decedent groups (deaths of despair [suicide, drug addiction, alcoholism], heart disease, and cancer) were descriptively compared using one-way analysis of variance. Subsequently, multinomial logistic regression analysis was conducted to estimate the log odds of death of despair, in contrast to death due to heart disease or death due to cancer, at various dimensions of psychological vulnerabilities. We examined psychological vulnerabilities × education interaction terms to examine whether the associations between psychological vulnerabilities and the risk of death of despair differed by education levels. The analysis was conducted for the gender-combined sample and gender-separate sample to examine gender differences in the associations between psychological vulnerabilities and relative risk of deaths of despair. Following the primary analyses, models were rerun with select health behaviors (binge drinking, smoking) as additional covariates to explore whether the risk of deaths due to despair relative to other categories of death were impacted by such health behaviors. Analyses were conducted using SPSS 27 (IBM, Armonk, NY, USA). 

## 3. Results

Table 1 presents descriptive statistics for decedents who died of despair-related causes (suicide, drug addiction, alcoholism), cancer, and heart disease. Members of the three cause of death groups differed in age, marital status, and employment status. Those who died of despair-related causes were the youngest of the three decedents groups at baseline and those who died due to heart disease were the oldest at baseline. In addition, decedents whose cause of death was despair-related were less likely to be married at baseline than those who died due to cancer. Decedents who died of heart disease were less likely to be working at baseline than decedents who died of other causes. Total household income differed across the cause of death groups at a trend level, with decedents who died of heart disease having the lowest household income at baseline. The three cause of death groups did not differ significantly in terms of gender, race/ethnicity, and education. 

Some aspects of psychological vulnerability at baseline differed significantly across the three cause of death groups. Specifically, decedents who died of despair-related causes reported lower levels of autonomy and higher levels of negative affect than those who died of cancer. In addition, there was a trend that those who died of despair-related causes reported lower purpose in life and lower environmental mastery than those who died due to cancer or heart disease. The three cause of death groups had comparable results for other psychological characteristics. 

Multinomial logistic regression models predicting cause of death via sociodemographic characteristics and psychological vulnerabilities, without interactions terms, revealed no statistically significant associations between psychological vulnerabilities and the relative risks of death of despair, death due to cancer, and death due to heart disease (results not shown).

Table 2 and Table 3 present a summary of the results of the multinomial logistic regression analysis predicting the relative risks of death of despair, cancer, and heart disease among those with deficiencies in eudaimonic well-being or hedonic well-being, contingent on decedents’ educational attainment (full results are presented in the Appendix A). The results in Table 2 show that low levels of eudaimonic well-being, including purpose in life, personal growth, and positive relations with others predicted a significantly greater risk of death of despair (relative to death due to heart disease) among decedents with a college education. Among decedents with a college degree, those whose purpose in life, personal growth, and positive relations with others were in the lowest tertile at baseline had a significantly greater risk of death of despair (relative to death due to heart disease) than their counterparts whose eudaimonic well-being scores were in the highest tertile at baseline. These findings are illustrated graphically in Figure 1, which also shows comparable levels of risk for deaths of despair (compared to heart disease) among those without a college degree across all levels of these aspects of well-being. 

Further, among college graduates, a low level of environmental mastery was a risk factor for death due to cancer relative to death due to heart disease. Specifically, decedents who had a college degree and reported environmental mastery in the lowest tertile had a greater risk of death due to cancer (compared to heart disease) than their peers who had environmental mastery scores in the highest tertile. Neither self-acceptance nor autonomy predicted the relative risk of death of despair. 

Table 3 shows that deficiencies in hedonic well-being, indicated by low levels of positive affect and life satisfaction and high levels of negative affect significantly predicted the relative risk of deaths of despair, deaths due to cancer, and deaths due to heart disease. Specifically, decedents with a college degree who scored the lowest tertile of positive affect at baseline had a greater risk of death of despair (relative to death due to heart disease) than their college-educated counterparts who had scores in the highest tertile of positive affect (also shown in Figure 1). 

Decedents whose life satisfaction at baseline was in the lowest tertile were more likely than those whose score was in the highest tertile to die of heart disease (relative to dying of cancer), if they had a college degree. Alternatively, decedents whose negative affect scores were in higher tertiles (middle or highest tertile) were more likely to die of cancer (relative to heart disease) than their counterparts whose negative affect scores were in the lowest tertile. 

Provisional analyses added two relevant health behaviors to the model (binge drinking, smoking) to see if they modified the above findings, possibly illuminating behavioral pathways toward particular kinds of death. These analyses showed no change in findings when binge drinking was added to the models. However, the additional control of smoking eliminated the prior findings for two outcomes. Specifically, low levels of positive relations with others and low levels of person growth were no longer significant predictors of increased risk of death of despairs, relative to the risk of death due to heart disease, when smoking was added to the models.

## 4. Discussion

The purpose of this investigation was to examine whether vulnerabilities in psychological well-being, eudaimonic or hedonic, increase risk for deaths of despair due to suicide, drug addictions, and alcoholism compared to risk of death due to heart disease or cancer. The questions were examined using mortality data from the MIDUS national longitudinal study. Those in the deaths of despair group were significantly younger at baseline than those who died from heart disease or cancer, thus, underscoring Case and Deaton’s [2] original emphasis on this kind of mortality happening in midlife. Those dying of despair were also less likely to be married at baseline compared to those who died of cancer. Alternatively, those who died of heart disease were less likely to be working at baseline compared to the other two decedent groups. Descriptive differences were not evident with regard to gender, race/ethnicity, or education.

Key findings from the multinomial logistic regressions were that deficiencies in reported levels of eudaimonic well-being, specifically, purpose in life, personal growth, and positive relations with others were linked with higher risk of deaths of despair compared to risk of death due to heart disease, after adjusting for sociodemographic covariates. Similarly, deficiencies in hedonic well-being, specifically, positive affect were also linked with higher risk of deaths of despair compared to risk of death due to heart disease. Thus, multiple types of psychological vulnerabilities increased risk for deaths of despair, thereby clarifying explicit meanings of despair in these types of death, which has been missing from prior studies.

Provisional analyses exploring the impacts of relevant health behaviors (binge drinking, smoking) examined possible behavioral pathways to specific type of deaths. It is noteworthy that those dying from suicide, addictions or alcoholism were more likely to smoke (58%) compared to those dying from heart disease (14%) or cancer (18%) at baseline. The results from analyses including smoking imply that such health behavior might be part of the behavioral pathways to despair-related deaths. Findings showed that psychological vulnerability was a predictor of smoking, particularly among those who had college degree. Further, including smoking in the models eliminated some significant associations between deficiencies of psychological well-being and relative risk of death of despair, thus pointing to a possible behavioral pathway. Future work is needed, however, to explicate mechanistic pathways to deaths of despair, which may implicate differing levels of stress exposures and lifestyle choices as well as differing profiles of coping strategies or personality traits. 

Importantly, all of the above findings were evident among adults with a college education compared to adults without a college degree. This finding appears contrary to the initial Case and Deaton’s work [2], which showed increased risk of deaths of despair among middle-aged whites of low socioeconomic status. However, subsequent studies have reported that premature mortality was not limited to middle-aged, uneducated, non-Hispanic white individuals [3]. Those endeavors have not included assessments of despair, however. Goldman, Glei, and Weinstein [6], in contrast, showed declining mental health over time, particularly for positive affect and life satisfaction (along with increments in negative affect) among low-SES adults across a broad range of ages, but their inquiry focused on those who were alive, not those who had died. 

We bring several observations to the issue of socioeconomic disadvantage, framed here in terms of educational status, to the understanding of deaths of despair. First, because psychological assessments of despair have been missing from most prior studies of premature mortality, it is largely unknown what psychological factors are involved increasing risk for certain kinds of death. Second, efforts to track eudaimonic and hedonic well-being at the population level provide useful information on historical change in psychological strengths and vulnerabilities over time, but they do not illuminate whether such factors increase risk for death, due to despair-related causes, or other causes. Findings from this inquiry thus bring psychological factors as important additions to mortality analyses, examined with national longitudinal data. 

Making sense of the heightened risk for deaths of despair compared to dying from heart disease *among college-educated adults* who lack key aspects of eudaimonic well-being—namely, purpose in life, personal growth, and positive relations with others, as well as a key aspect of hedonic well-being, namely, positive affect, requires consideration of the social structural contouring of positive mental health. It is well documented that those with higher educational and economic standing tend to have higher levels of eudaimonic well-being compared to their less privileged counterparts [34,35,36,37]. Similarly, hedonic well-being is known to be positively linked with economic and educational advantage [37,38,39]. Such realities underscore that among privileged segments of society, it is something of an anomaly to *not experience well-being*.

Such deficiencies may heighten risk for other factors that increase risk of despair-related mortality. For example, Shanahan et al. [16] describe cognitive biases that may deepen and perpetuate negative life outlooks as well as reckless, risky, and unhealthy behaviors, both of which may fuel emotional distress (sadness, anhedonia, apathy). Thus, the work reported herein points to further scientific directions to increase understanding of mechanistic pathways leading to deaths due to suicide, alcoholism, and drug addiction. 

A key question is how diverse vulnerabilities are distributed across the socioeconomic hierarchy. Perhaps among less educated, less privileged members of society, there is heightened risk of multiple adversity factors that increase risk for deaths of despair. As illustrated in the figures accompanying key findings, among those with less than a college education, there was comparably high risk of despair-related deaths, regardless of level of well-being, whereas it was only those in the lowest tertile of well-being who had high risk among the college educated. Such contrasting patterns call for additional inquiries that incorporate, not only cognitive, behavioral, and emotional risk factors, as noted above, but also stress exposures among those who die from despair-related deaths. Perhaps different combinations of risk factors account for such deaths among advantaged vs. disadvantaged segments of society.

The MIDUS study includes extensive prior research on biopsychosocial pathways to mortality. For example, financial stress during the Great Recession has been linked to early mortality [40] while perceived control has been found to reduce mortality risk at low but not high levels of education [41], and SES combined with depressive affect and diurnal cortisol predicts all-cause mortality [42]. Other findings have linked changes in happiness to physical health and mortality [43]; well-being has also been combined with residential mobility to predict mortality [44]. Numerous studies have brought social relational experience, such as giving and getting [45], social support and strain [46], partner responsiveness [47,48], and loneliness and social isolation [49] to all-cause mortality. Other work has focused on specific experiences, such as childhood adversity combined with diabetes [50], or death of a child [51], or perceived discrimination [52,53], or daily stress combined with glycemic control [54] to predict mortality. Finally, health behaviors, such as smoking and alcohol/drug abuse [55] and variation in sleep duration [56] also predict mortality.

The point of highlighting wide-ranging interest in factors that increase risk of death in MIDUS is to underscore that these prior inquiries routinely use survival analyses focused on *all-cause mortality*. That is, specific causes of death are not examined, and the samples include both those who are still living and those who have died. Based on findings in this analysis, opportunities exist for future inquiries to investigate risk factors for specific causes of death among decedents. Here, we found that low levels of multiple aspects of well-being created heightened risk for despair-related deaths compared to deaths from heart disease. The broader implication is that many of the previous all-cause mortality inquiries could be enriched by formulating and testing cause-specific pathways to different kinds of death. 

That deficiencies in widely studied aspects of well-being, both eudaimonic and hedonic, increased risk of death due to suicide, drug addictions, and alcoholism compared to deaths due to heart disease has implications for intervention. Prior research documents that well-being is modifiable and can be improved [57,58], including work documenting benefits of well-being training among older adults [59], along with meta-analyses of randomized control trials showing that interventions can increase psychological well-being [60]. Community-based interventions with older adults have also shown that multiple aspects of eudaimonic well-being can be enhanced and psychological distress can be reduced [61]. Stated otherwise, the psychological vulnerabilities that increase risk of deaths of despair can become targets for public health education and specific intervention strategies. It is noteworthy that increased deaths of despair are mainly attributable to a substantial increase in drug-related deaths [62], thus underscoring the importance of such interventions in the context of the opioid epidemic. 

Some limitations of this study should be acknowledged. The small number of deaths in specific cause of death groups, particularly for analyses with interactions between levels of psychological vulnerabilities (i.e., three groups based on tertile values) and education (i.e., two groups depending on bachelor’s degree attainment), raises concern about Type II errors. Although the present study is a first step in this line of research, testing models with a larger number of deaths will strengthen the findings and reinforce their implications. Future studies with a larger sample would also allow consideration of additional socioeconomic factors such as race/ethnicity that were controlled for in the present analyses. The current study has a potential for such expansion by combining mortality data from MIDUS Refresher (2012), an additional cohort of respondents, with data from the core sample. Another limitation relates to cause of death information in ICD-9 or ICD-10 formats. A recent study indicated that over 30% of deaths are garbage coded, i.e., cause of death in ICD codes were incorrectly or vaguely assigned and consequently mask the true cause of death distribution [63]. This suggests that a portion of deaths of despair cases might have been excluded from the current inquiry due to limitation in the cause of death information in ICD codes. The implication is that our findings likely underestimate the true effect. 

## 5. Conclusions

The present findings advance scientific understanding of deaths of despair by identifying specific psychological deficiencies that increase risk for these types of mortality. Data from the MIDUS national longitudinal study were used to investigate these questions. Because well-being is modifiable, the obtained results point to important directions for future work to enhance experiences of purpose in life, personal growth, positive relations with others, and positive affect among those who do not experience such types of well-being, in hopes of promoting longevity. Further efforts to enrich extensive all-cause mortality analyses in MIDUS with new inquiries focused on risk pathways to specific causes of death were delineated.

## Figures and Tables

**Figure 1 ijerph-20-06480-f001:**
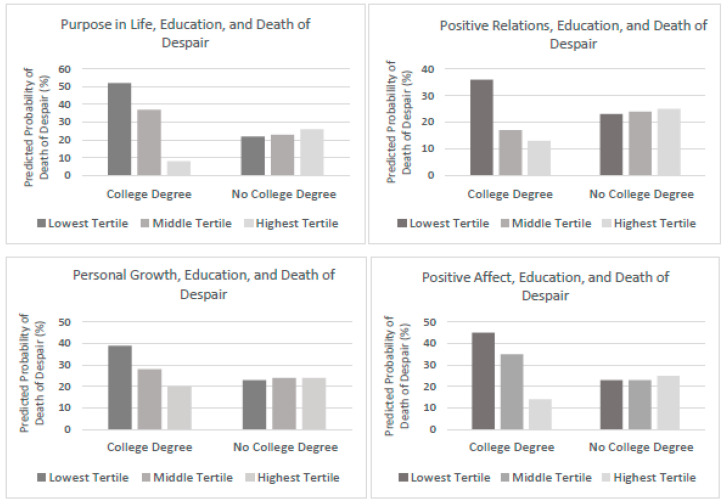
Predicted Probability of Deaths of Despair, relative to deaths of heart disease, by Levels of Eudaimonic and Hedonic Well-Being and Education in MIDUS 2 Participants.

**Table 1 ijerph-20-06480-t001:** Descriptive Statistics for Decedents by Cause of Death at MIDUS 2 (2004–2006).

	Death of Despair	Death of Cancer	Death of Heart Disease	*p*
	M (SD) or %	M (SD) or %	M (SD) or %	
Age	61.1 (11.2)	64.5 (10.7)	69.0 (10.0)	***
Women, %	44.1	50.4	46.5	ns
Non-Hispanic white, %	76.1	77.7	81.5	ns
College degree, %	22.3	30.0	25.5	ns
Household income	49,469 (50,361)	53,257 (48,330)	43,819 (48,078)	+
Married, %	51.5	66.4	59.8	**
Working, %	46.7	46.9	32.1	***
Eudaimonic well-being				
Purpose in life	35.8 (7.7)	37.5 (6.8)	36.6 (7.6)	+
Positive relations with others	39.5 (7.5)	40.9 (7.2)	40.0 (7.4)	ns
Personal growth	36.7 (7.8)	37.7 (7.3)	36.8 (7.2)	ns
Environmental mastery	37.1 (7.9)	38.8 (7.3)	38.1 (7.4)	+
Self-acceptance	36.9 (8.8)	38.4 (7.7)	38.0 (7.8)	ns
Autonomy	36.7 (7.2)	38.5 (6.7)	37.9 (6.6)	*
Hedonic well-being				
Positive affect	3.51 (0.8)	3.60 (0.8)	3.59 (0.8)	ns
Negative affect	1.68 (0.7)	1.54 (0.6)	1.52 (0.6)	*
Life satisfaction	7.44 (1.6)	7.64 (1.3)	7.61 (1.5)	ns
*N*	168	256	271	

Note: Causes of death (death of despair, death of cancer and death of heart disease) were identified based on multiple cause of death and underlying cause of death information from NDI Plus (2004–2022). Death of despair includes death from suicide, drug addiction, and alcoholism. *** *p* < 0.001, ** *p* < 0.01, * *p* < 0.05, + *p* < 0.10. ns = non-significant.

**Table 2 ijerph-20-06480-t002:** Summary of Multinomial Logistic Regressions Predicting Different Types of Death by Levels of Eudaimonic Well-Being, Education, and Gender.

	Findings	Full Sample	Men	Women
Purpose in life (PL)	Higher risk of death of despair (vs. heart disease) in presence of the lowest tertile PL, among decedents with college degree (vs. no college degree)	**	+	+

Positive relations with others (PR)	Higher risk of death of despair (vs. heart disease) in presence of the lowest tertile PR, among decedents with college degree (vs. no college degree)	*	ns	+

Personal growth (PG)	Higher risk of death of despair (vs. heart disease) in presence of the lowest tertile PG, among decedents with college degree (vs. no college degree)	*	ns	+

Environmental mastery (EM)	Higher risk of death due to cancer (vs. heart disease) in presence of the lowest tertile EM, among decedents with college degree (vs. no college degree)	*	**	ns

Self-acceptance (SA)	Higher risk of death due to cancer (vs. heart disease) in presence of the lowest tertile SA, among male decedents with college degree (vs. no college degree) at trend level	ns	+	ns

Autonomy (AU)	No significant associations	ns	ns	ns

Note: Causes of death (death of despair, death of cancer and death of heart disease) were categorized based on multiple cause of death and underlying cause of death information from NDI Plus (2004–2022). Death of despair includes death from suicide, drug addiction, and alcoholism. All models were adjusted for age, race/ethnicity, gender (for full sample), education, household income, marital status, and employment status at MIDUS 2 baseline (2004–2006). ** *p* < 0.01, * *p* < 0.05, + *p* < 0.10. ns = non-significant.

**Table 3 ijerph-20-06480-t003:** Summary of Multinomial Logistic Regressions Predicting Different Types of Death by Levels of Hedonic Well-Being, Education, and Gender.

	Findings	Full Sample	Men	Women
Positive affect	Higher risk of death of despair (vs. heart disease) in presence of the lowest tertile positive affect, among decedents with college degree (vs. no college degree)	* lowest tertile (vs. highest tertile)	ns	ns

Negative affect	Higher risk of death due to cancer (vs. heart disease) in presence of higher tertiles (middle, highest) negative affect, among decedents with college degree (vs. no college degree)	** middle tertile * highest tertile (vs. lowest tertile)	** middle tertile + highest tertile (vs. lowest tertile)	ns

Life satisfaction	Higher risk of death of despair (vs. heart disease) in presence of the lowest tertile life satisfaction among women decedents with college degree (vs. no college degree); Higher risk of death of heart disease (vs. cancer) in presence of lower tertiles (middle, lowest) life satisfaction among decedents with college degree (vs. no college degree)	+ D of cancer at middle tertile (vs. highest tertile)	ns	+ D of despair * D of cancer at lowest tertile (vs. highest tertile)

Note: Causes of death (death of despair, death of cancer, and death of heart disease) were identified based on multiple cause of death and underlying cause of death information from NDI Plus (2004–2022). Death of despair includes death from suicide, alcoholism, and drug addiction. All models were adjusted for age, gender (for full sample), race/ethnicity, education, household income, marital status, and employment status at MIDUS 2 baseline (2004–2006). ** *p* < 0.01, * *p* < 0.05, + *p* < 0.10. ns = non-significant.

## Data Availability

The data is available via ICPSR (Inter-university Consortium for Political and Social Research): https://www.icpsr.umich.edu/web/ICPSR/series/203 (accessed on 27 April 2023).

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
