# Peer review of "Unpacking Psychological Vulnerabilities in Deaths of Despair"

_ijerph, 2023, doi:10.3390/ijerph20156480_

Round 1

Reviewer 1 Report

Dear Authors, thank you for your interesting study which addresses important issues on effects of well-being in broad context. I have just few minor comments:

1. Lines 130-132: it would be great if you reformulate this sentence. In the beginning it feels like 695 participant died from despair, but by the end of the sentence we find out, that it was a total number.

2. There is no explanation for NDI plus. What is that?

Author Response

  1. Lines 130-132: it would be great if you reformulate this sentence. In the beginning it feels like 695 participant died from despair, but by the end of the sentence we find out, that it was a total number.

Response: The sentence has been revised as follows:

“The analytic sample consisted of the decedents who died of despair related causes (suicide, drug addiction, alcoholism) (n = 168; 94 men, 74 women), cancer (n = 256; 127 men, 129 women), or heart disease (n = 271; 145 men, 126 women).   

  1. There is no explanation for NDI plus. What is that?

Response: NDI Plus (National Death Index Plus) provides the cause of death information in addition to basic NDI information (NDI User’s Guide: https://www.cdc.gov/nchs/data/ndi/ndi_users_guide.pdf). It is added to the lines 127-129.

Reviewer 2 Report

This study employs longitudinal data from the Midlife in the U.S. (MIDUS) study to examine whether vulnerabilities in eudaimonic and hedonic well-being, increase the risk of death of despair compared to the risk of death due to heart disease or cancer. One of the main contributions of this paper is its focus on psychological vulnerabilities as a potential explanation for deaths of despair. By examining both eudaimonic and hedonic well-being, the authors provide a nuanced understanding of how different aspects of psychological well-being may contribute to these deaths. Additionally, by using longitudinal data from MIDUS, they are able to examine these vulnerabilities over time and provide insight into how they may develop and change. There are, however, some issues that could be addressed to strengthen the contributions of this paper.

In general, the introduction section is well-written. It would be great if the authors could rearrange your introduction by removing sections that aren't directly connected to your present research question. For example, you mentioned the cohort trend in psychological distress, however, this study does not specifically address the cohort implications of psychological distress on mortality risks. The paragraph starting from line 81 is also peripheral regarding how life's purpose originated and evolved. Please consider removing this information.  

In the method section, I would like to see if controlling for health behaviors, e.g., smoking, drinking, etc., could change the current findings, since those are significant confounders in both mortality and related disease risks if these details are available in this data. 

Regarding Tables 2 and 3, I noticed that the authors report the main findings but do not include the coefficients. It can be helpful to present estimated coefficients in tables to help readers comprehend the magnitude of the effects, or at least present entire models as supplemental materials.

Furthermore, because the interpretations of multinomial logit models are not always intuitive, I recommend that the authors present the marginal effect of multinomial logit regression, which has more straightforward interpretations.

A minor point to mention is that the underlying social processes for the death of despair and other types of death are likely to differ. While heart disease and cancer may be linked to aging, lifestyle choices, such as diet and exercise, deaths from suicide, drug addiction, and alcoholism are often rooted in deeper emotional and psychological issues. It calls into question the rationale for comparing these different types of deaths and the factors that contribute to them. It can be expected that psychological well-being may be less predictive of heart disease mortality and cancer compared to the death of despair. 

By restricting your sample to only include deceased participants, you run the risk of leaving out healthier groups and missing the opportunity to include a healthy control. In light of this, I also wonder if you could perform a survival analysis and possibly incorporate a competing risk component into the survival model to estimate the impact of eudaimonic and hedonic well-being on cause-specific mortality risks and prevent potential selection bias.

The finding that positive relations with others predict a higher likelihood of death of despair is a little bit surprising; could you provide some insights into why this is the case? Could you provide some discussion on how educational effects moderate the effects of psychological well-being and the death of despair, given the vastly distinct patterns between higher education and lower education groups presented in the figures? You would benefit from a more thorough explanation of how stress operates in the process to unpack the mechanism linking psychological well-being and the death of despair.   

The finding that positive relationships with others predict a higher likelihood of death from despair is a little surprising; could you explain why this is so? Could you provide more discussion on how educational effects moderate the effects of psychological well-being and death of despair, given the vastly different patterns shown in the figures for higher education and lower education groups? Stress, as you mentioned briefly, can play a role in shaping the education effects on these outcomes.  A more detailed explanation would help to unpack the mechanism linking psychological well-being and the death of despair.

This discussion of biopsychosocial pathways to mortality is somewhat general and does not directly distinguish between types of death; however, since distinguishing between different types of death is the main contribution of your study, you may consider rephrasing this section so that these possible mechanisms can speak directly to your findings.

Author Response

In general, the introduction section is well-written. It would be great if the authors could rearrange your introduction by removing sections that aren't directly connected to your present research question. For example, you mentioned the cohort trend in psychological distress, however, this study does not specifically address the cohort implications of psychological distress on mortality risks. The paragraph starting from line 81 is also peripheral regarding how life's purpose originated and evolved. Please consider removing this information.  

Response: Thank you for this input as it helps us clarify why our introduction covered the cohort trend in psychological distress.  First, the phenomenon of deaths of despair emerged over time – it was first identified in 2015 by Case & Deaton and has subsequently been prominent in many inquiries.  Relatedly, findings from MIDUS, which include two national samples moving through time, one initiated in 1995 and another in 2012, document declines in mental health for more recent cohorts, particularly those of lower socioeconomic standing.  We believe these temporal findings are central to understanding why deaths of despair have become of interest in numerous scientific studies, including MIDUS. 

The text about purpose in life is critical for thinking about how to approach the phenomenon of despair, which has been poorly formulated thus far, conceptually and empirically.  Victor Frankl’s powerful message about purpose in life being life-sustaining in the face of monumental adversity (3 years in Nazi concentration camps) offers a useful framework for scientific studies of despair – defined as low purpose in life.

Case, A., & Deaton, A. (2015). Rising morbidity and mortality in midlife among white non-Hispanic Americans in the 21st century. Proceedings of the National Academy of Sciences of the United States of America112(49), 15078–15083. https://doi.org/10.1073/pnas.1518393112

In the method section, I would like to see if controlling for health behaviors, e.g., smoking, drinking, etc., could change the current findings, since those are significant confounders in both mortality and related disease risks if these details are available in this data. 

Response: Per the reviewer’s suggestion, we ran the models with additional controls for relevant health behaviors (smoking and binge drinking) at baseline (MIDUS2).  That is to say, such factors may be construed, not as confounders, but possibly as part of the behavioral pathway toward particular kinds of death.  The results showed that the additional control of ‘binge drinking’ did not change the findings.  However, models with additional control of ‘smoking’ eliminated some significant associations between lack of psychological well-being (specifically, positive relations with others and personal growth) and risk of deaths of despair compared to death of heart disease.  We now include these additional health behaviors in our reported findings, as possible pathways rather than confounders, while also underscoring notable differences in rates of smoking between those who died of despair related causes (58%) compared to deaths due to heart disease (14%) or cancer (18%). Relatedly, the results from these exploratory analyses imply that health behaviors might be part of the behavioral pathways to despair-related deaths.   The logistic regression analysis predicting the probability of smoking showed that deficiency of psychological well-being (e.g., the lowest tertile purpose in life) was a predictor of smoking, particularly among those who had college degree, which aligns with prior findings in the current paper. Including smoking in the models eliminated some significant associations between deficiencies of psychological well-being and relative risk of death of despair, which could be interpreted as a behavioral pathway.  That said, we emphasize in findings and the Discussion that explicating the mechanisms/pathways, which lie behind deaths of despair, is beyond the focus of the current paper.  Instead, we frame those topics as important directions for future research.

Regarding Tables 2 and 3, I noticed that the authors report the main findings but do not include the coefficients. It can be helpful to present estimated coefficients in tables to help readers comprehend the magnitude of the effects, or at least present entire models as supplemental materials.

Response: Per the reviewer’s suggestion, we added the entire models as supplemental materials.

Furthermore, because the interpretations of multinomial logit models are not always intuitive, I recommend that the authors present the marginal effect of multinomial logit regression, which has more straightforward interpretations.

Response: Per the reviewer’s suggestion, marginal effects and probabilities of death of despair in various subgroups were estimated for straightforward interpretation of the findings. We agree that this approach improves interpretability of the results and thus now include a modified Figure 1 that illustrates the predicted probability of deaths of despair.

A minor point to mention is that the underlying social processes for the death of despair and other types of death are likely to differ. While heart disease and cancer may be linked to aging, lifestyle choices, such as diet and exercise, deaths from suicide, drug addiction, and alcoholism are often rooted in deeper emotional and psychological issues. It calls into question the rationale for comparing these different types of deaths and the factors that contribute to them. It can be expected that psychological well-being may be less predictive of heart disease mortality and cancer compared to the death of despair. 

Response: We do not believe sufficient evidence exists to assert that the underlying social processes for deaths of despair compared to other kinds of death are likely to differ.  Those are important future directions that involve lifestyle choices, socioeconomic standing, stress exposures – all possibly on the pathway to different kinds of death.  However, our objective in this initial inquiry was to assess deaths of despair as an empirical question that requires operationalizing possible meanings of despair (low levels of various kinds of well-being) and then assessing whether these psychological vulnerabilities were more consequential in predicting deaths due to suicide, addictions, alcoholism compared to deaths due to cancer or heart disease.  Indeed, our guiding hypothesis follows from the reviewer’s observation: namely that psychological ill-being would be less predictive of heart disease and cancer compared to deaths due to suicide, addictions, and alcoholism.  That is the empirical question motivating this inquiry.

By restricting your sample to only include deceased participants, you run the risk of leaving out healthier groups and missing the opportunity to include a healthy control. In light of this, I also wonder if you could perform a survival analysis and possibly incorporate a competing risk component into the survival model to estimate the impact of eudaimonic and hedonic well-being on cause-specific mortality risks and prevent potential selection bias.

Response:  We appreciate the suggestion to conduct survival analyses, but underscore that our central question in these analyses are not about predicting risk of mortality.  We note many such prior analyses from MIDUS have examined risk for all-cause mortality.  These are briefly showcased in our Discussion, where we make the point that such widely investigated survival analyses ask fundamentally different question than what is of central interest in this query, which is explicitly designed to investigate psychological vulnerabilities vis-à-vis different kinds of death.  That is, our key question is whether psychological despair (operationalized with various indicators) more strongly predicts deaths due to suicide, addictions, and alcoholism compared to deaths due to heart disease and cancer.  In large national longitudinal studies such as MIDUS, much is to be learned not only from widely used survival analyses that include living respondents and those who have died from all causes, but also from comparative analyses among only decedents who died of different causes.  The former (all-cause mortality) inquiries are notably useful in identifying preventive/protective factors, while the latter offer insight into vulnerability factors that may heighten risk of certain kinds of death.

The finding that positive relations with others predict a higher likelihood of death of despair is a little bit surprising; could you provide some insights into why this is the case?

Response: The reviewer’s interpretation is incorrect. The finding related to the ‘positive relations with others (PR)’ actually showed ‘higher risk of death of despair, compared to heart disease, in presence of the lowest tertile PR’ (see Table 2).

Could you provide more discussion on how educational effects moderate the effects of psychological well-being and death of despair, given the vastly different patterns shown in the figures for higher education and lower education groups? Stress, as you mentioned briefly, can play a role in shaping the education effects on these outcomes.  A more detailed explanation would help to unpack the mechanism linking psychological well-being and the death of despair.

This discussion of biopsychosocial pathways to mortality is somewhat general and does not directly distinguish between types of death; however, since distinguishing between different types of death is the main contribution of your study, you may consider rephrasing this section so that these possible mechanisms can speak directly to your findings.

Response:  Again, the central purpose of this inquiry is not to investigate mechanistic processes that may be part of understanding how deaths of despair come about.  Rather, the key objective was to assess whether despair, operationalized as low levels of eudaimonic and hedonic well-being, actually predicts increased risk of death due to suicide, drug addictions or alcoholism compared to deaths due to heart disease or cancer.  We now add provisional analyses showing the assessments of smoking (but not binge drinking) impact some of our obtained outcomes, and we discuss what this might mean as well as other directions for future mechanistic inquiry.

Regarding the unexpected finding that our key predictions were more strongly evident among those with higher rather than lower educational standing, we offer several thoughts.  First, we note that although deaths of despair were initially thought to be evident primarily among those of lower socioeconomic status, additional findings now clarify that premature mortality has not been limited to middle-aged, uneducated whites.  We also reiterate that because psychological despair has been mostly inferred rather than actually measured in this literature, it is largely unknown what psychological factors best capture the idea of despair.  Perhaps most importantly, we remind readers that eudaimonic well-being is known to be higher among those with higher educational standing.  Thus, not experiencing these kinds of well-being if one is college educated may constitute a particular kind of deficiency that may deepen or perpetuate other cognitive biases.  These might contribute to reckless, risky behaviors that fuel emotional distress.  Stated otherwise, we point to possibly complex mechanistic pathways that may vary by educational status, which constitute worthwhile directions for future research. 

Round 2

Reviewer 1 Report

Thank you for your clarifications 

Reviewer 2 Report

The authors have addressed most of my previous concerns and improved the paper as a result.

English language fine.